# Detection of motor-related mu rhythm desynchronization by ear EEG

**Masaya Ueda**[1]*, **Keita Ueno**[1], **Takao Inoue**[1], **Misao Sakiyama**[1], **China Shiroma**[1,2], **Ryouhei Ishii**[1,3], **Yasuo Naito**[1]

1 Department of Occupational Therapy, Graduate School of Rehabilitation Science, Osaka Metropolitan University, Osaka, Japan, 2 Rehabilitation Unit, Murata Hospital, Osaka, Japan, 3 Department of Psychiatry, Graduate School of Medicine, Osaka University, Osaka, Japan

* uedam@omu.ac.jp

## Abstract

Event-related desynchronization (ERD) of the mu rhythm (8–13 Hz) is an important indicator of motor execution, neurofeedback, and brain-computer interface in EEG. This study investigated the feasibility of an ear electroencephalography (EEG) device monitoring mu-ERD during hand grasp and release movements. The EEG data of the right hand movement and the eye opened resting condition were measured with an ear EEG device. We calculated and compared mu rhythm power and time-frequency data from 20 healthy participants during right hand movement and eye opened resting. Our results showed a significant difference of mean mu rhythm power between the eye opened rest condition and the right hand movement condition and significant suppression in the 9–12.5 Hz frequency band in the time-frequency data. These results support the utility of ear EEG in detecting motor activity-related mu-ERD. Ear EEG could be instrumental in refining rehabilitation strategies by providing in-situ assessment of motor function and tailored feedback.

## Introduction

In neurorehabilitation, finding metrics that are accurate and user-friendly for assessing motor function and recovery is critical, particularly in stroke rehabilitation. Electroencephalography (EEG) is a leading method for non-invasive measurement of human brain activity. Typical examples are the event-related desynchronization (ERD) of the mu rhythm (8-13 Hz) serving as an important indicator of motor execution and planning [1–6]. Mu-ERD is not only associated with recovery from stroke-induced motor disorder but has also been established as an indicator of neurofeedback and brain-computer interface [7–10]. While traditional scalp EEG setups can capture cortical activity with fine temporal precision, they often have spatial limitations due to electrode placement and are not suitable for everyday rehabilitation practice due to their laboratory-bound nature [9,11,12].

The field has witnessed a shift towards portable and wearable EEG systems due to technological advancements, making brain activity monitoring feasible in everyday settings [13–15]. However, most nonclinical EEG devices are designed for general-use EEG applications and lack support for sophisticated signal processing and effective feedback generation [16].

**Data availability statement:** All EEG data or demographic data are available from the Dryad database (accession URL: http://datadryad.org/stash/share/UZfy8id2Y9zB0VPLsEkDRTZXdrlqXAFd81RXhrtwTxw).

**Funding:** Initials of the authors who received each award: M.U Grant numbers awarded to each author: 23K16581 and 22K21212 The full name of each funder: Japan Society for the Promotion of Science(JSPS) URL of each funder website: https://www.jsps.go.jp. The funders had no role in study design, data collection and analysis, decision to publish, or preparation of the manuscript.

**Competing interests:** The authors declare that they have no known competing financial interests or personal relationships that could have appeared to influence the work reported in this paper.

The use of wearable EEG devices has been limited primarily to frontal scalp measurements, providing a narrow view of motor-related brain activity [11,17]. Ear EEG is a new approach that holds the potential to enhance the feasibility of long-term observation of brain activity in clinical settings [18–20]. Ear EEG has been validated using common EEG paradigms such as alpha attenuation and auditory evoked potentials and is able to accurately characterize the frequency spectrum and has similar signal-to-noise ratios as nearby scalp electrodes [13,19]. In addition, previous studies have shown a high correlation between the ear EEG and electrodes located in the temporal and inferior parietal lobe [19,21,22]. Nevertheless, the efficacy of ear EEG in recording motor-related brain dynamics, particularly mu-ERD, during specific motor tasks is not yet fully established. This represents an important area for further research to validate the ability of ear EEG to identify motor-related neural signatures, such as mu-ERD, across different motor tasks, potentially expanding the application of EEG in neurorehabilitation [12,23].

Our study examined the feasibility of using an ear EEG device to monitor the ERD of the mu rhythm during the hand movement. We analysed the EEG data of the right hand movement and the following eye opened resting condition were measured with an ear EEG device. Confirmation of its utility would validate ear EEG as a tool for assessing motor-related brain activity, providing an accessible and less invasive alternative to traditional EEG setups. It would also underscore the potential of ear EEG to support the development of novel, personalized rehabilitation approaches.

## Methods

### Participants

This study involved 27 healthy volunteers (mean age: 22.6 ± 4.6 years, six males). Minors were not included in the participants. Participants were recruited from August 1, 2023, to January 31, 2024. The dominant hand was evaluated in FLANDERS Handedness Questionnaire Japanese. Before recruiting, informed consent was obtained in writing from all participants included in the study. This study was conducted with the approval of the Osaka Prefecture University Graduate School General Rehabilitation Studies Ethics Committee (2023–211). All the procedures were conducted following the tenets of the Declaration of Helsinki.

### EEG device and setup

We used XHOLOS Sounds Ear EEG Device (CyberneX Corporation, Japan) for measuring EEG. This device features a unique 1-channel system. The measurement electrode is the right ear canal, and the reference and ground electrodes are the left ear canal using ear-tips (Ear-tip type flexible electrode, base material: Si rubber) as electrodes (Fig 1). It has a sampling rate of 600 Hz and meets the Japanese Industrial Standards (JIS, T 1203) for medical EEG equipment in terms of signal quality, impedance, and noise level. EEG signals were transmitted via Bluetooth to the measurement laptop PC by HP 16-wf0000 (HP Corporation, Japan) and recorded using the measurement application (CyberneX Corporation, Japan). The electrode was sat in both ear canals after being wiped with an alcohol swab. The sampling rate was set to 600 Hz.

### Experimental procedure

The EEG measurements were performed in the relaxed sitting position. All participants completed eye closed resting, eye opened resting, and right-hand movement condition. The alpha-blocking, which an open/closed-eye task can confirm, was used to verify the accuracy of EEG measurements. Below is an overview of the tasks (Fig 2). EEG signals were recorded under the following three conditions.

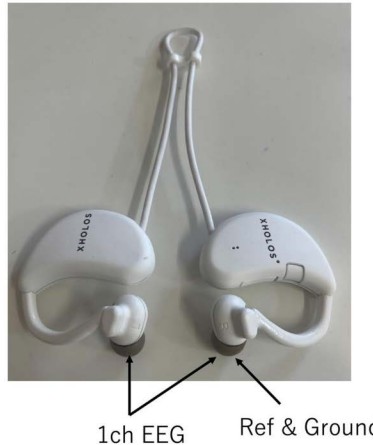
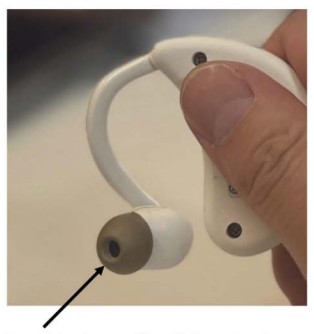

1ch EEG   Ref & Ground'   Ear-tip type flexible electrode

**Fig 1. Photograph of the ear EEG device and ear-tip.** This ear EEG device features a unique 1-channel system. The measurement electrode is the right ear canal, and the reference and ground electrodes are the left ear canal using ear tips (Ear-tip type flexible electrode, base material: Si rubber) as electrodes.

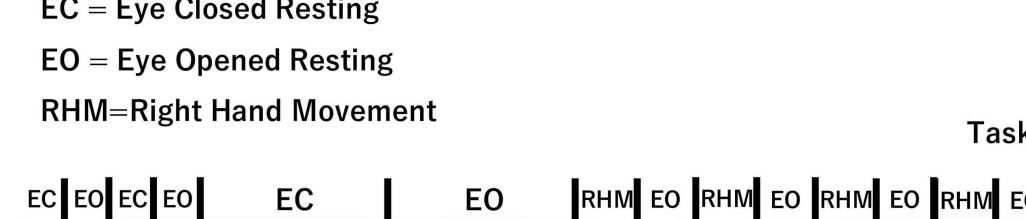
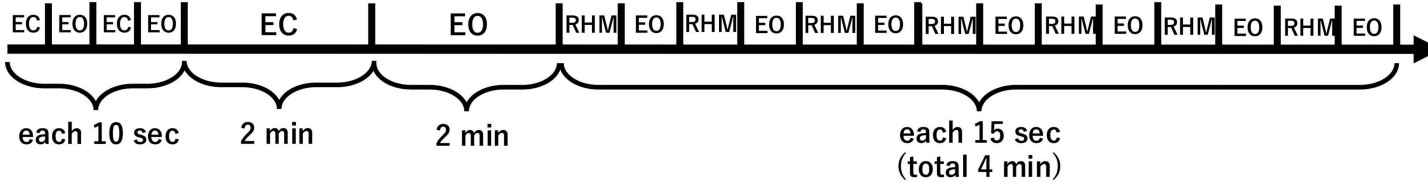

**Fig 2. Overview of the experimental task for EEG measurement.**

**Eye closed resting condition.** Participants were instructed to close their eyes while sitting on a chair. The eyes were then kept closed for 2 min.

**Eye opened resting condition.** Participants were instructed to open their eyes while sitting in a chair and to look at a fixation point on the table in front of them while relaxing for 2 minutes.

**Right-hand movement condition.** Participants were instructed to perform 15 seconds of repetitive right-hand grasp and release movements with eyes opened, followed by 15 seconds of eye opened resting. This sequence was repeated eight times during the session. The hand-grasp and release movements were designed to simulate functional tasks to ensure the relevance of the EEG data to motor activities. The timing of the start and end of the hand movement was confirmed by video recordings synchronized with the EEG. We chose the right-hand movement because most of the subjects were right-handed. Previous studies show that handedness influences the proportion of pre-movement mu-ERD in the left and right peri-rolandic areas [24,25].

## EEG data acquisition and analysis

EEG data obtained from 27 participants underwent thorough visual inspection and assessment for the presence of alpha-blocking phenomenon and overall signal quality by a physician certified by the Japanese Society of Clinical Neurophysiology.

The EEG data from the right hand movement condition and the subsequent eye open resting period were analyzed using Brain Electrical Source Analysis: BESA Research 7.1 software developed by BESA GmbH, Germany. As part of the offline processing, a high-pass filter set at 4.0 Hz and a low-pass filter set at 40.0 Hz were applied. The EEG data from each condition were divided into 2-second segments for statistical analysis, and the epoch-included potential artifacts, such as blinks and electromyograms, were visually eliminated, while further artifact detection scans were performed with an amplitude threshold set at $100\,\mu\mathrm{V}$. The average number of remaining epochs during the eye opened resting condition was calculated to be 54 ± 2, and for the right hand movement condition, the average number was 52 ± 4. First, analyses with Fourier transforms were computed using a multi-taper method for each condition [26,27] at a 0.25 Hz resolution. This resulted in estimates of absolute spectral power sampled for every 1 Hz bin during the interval of 4–40 Hz. Second, the brain oscillatory activity changes during right hand movement condition and eye opened resting condition were transformed into the time-frequency domain by using complex demodulation [28,29] (for detailed information on this methodology, see reference No.26,27). The frequency spectrum ranged from 4.0 Hz to 30.0 Hz in 0.5 Hz increments with a time sampling rate of 100 ms. After that, mu rhythm power (8-13 Hz) and the time-frequency data of the right hand movement condition followed by the eye open rest condition were analyzed for each participant and a grand average was made for all participants.

## Statistical analysis

To compare the averaged mu rhythm power (8-13 Hz) of each participant between the right hand movement condition versus the eye open rest condition, Wilcoxon signed rank test analyses were performed using SPSS software version 28. The significance level was set at 0.05.

Statistical differences of time-frequency data between the right hand movement condition and the open eye rest condition were analyzed by permutation test based on t-test and cluster analysis using BESA Statistics 2.1 (BESA GmbH, Germany). We set the average over time to average the 2-second windows, and a paired t-test with 1000 permutations was used to compare the EEG data of each condition. When comparing the differences in time-frequency data, a comparison is made for each voxel that is output according to the time-frequency resolution. Therefore, it is necessary to correct for multiplicity. BESA Statistics uses parameter-free permutation testing based on the Student's t-test [30,31]. In this study, there were no predefined clusters because BESA Statistics automatically identifies clusters in time and frequency that are significantly different between the two conditions. The null hypothesis of "the data under the experimental conditions comes from the same probability distribution" was rejected if at least one t-value was above the critical threshold for $p < 0.05$ determined by 1000 permutation.

## Results

### Participants and signal quality EEG data

We recruited 27 healthy volunteers (mean age: 22.6 ± 4.6 years, 6 males). However, 7 of the 27 participants did not complete the measurement protocol because the size of the ear tips did not fit the ear canal. Finally, 20 participants (mean age: 22.9 ± 4.9 years, 6 males) were included in the analysis. Results of the FLANDERS Handedness Questionnaire Japanese: 2 participants were left-handed.

In the EEG data of the 20 participants analysed, alpha-blocking was visually verified in the closed-eye condition. However, mu rhythm suppression could not be visually confirmed.

## Mu rhythm (8–13Hz) power

The time course of the grand average for the mu rhythm power is shown in Fig 3 and the mean mu rhythm power (μV²) for each participant in the right-hand movement condition and eye opened resting condition is shown in Fig 4. The mean and standard deviation of mu rhythm power in the eye opened resting condition was 5.46 ± 2.44 μV² and in the right hand movement condition was 4.92 ± 2.18 μV². The Wilcoxon signed rank test revealed a significant difference in the averaged mu rhythm power between each condition (P < 0.001).

## Time-frequency data

The grand average of the time-frequency data for 15 seconds of the eye opened resting condition and the right hand movement condition was showed Fig 5. The paired t-test with 1000 permutations for the time-frequency data showed a significant suppression in the 9-12.5 Hz frequency band during the right hand movements condition compared to the eye opened resting condition (p < 0.001) (Table 1). Interestingly, other significant reductions were also found at 4-6 Hz and 25-25.5 Hz (p = 0.001).

## Discussion

Our investigation examined the ability of the ear EEG device to monitor mu-ERDD during hand movement. While visual confirmation of mu rhythm suppression on the waveform was elusive, statistical analysis showed a significant difference in mu rhythm power between the eye opened rest condition and the right hand movement condition and significant

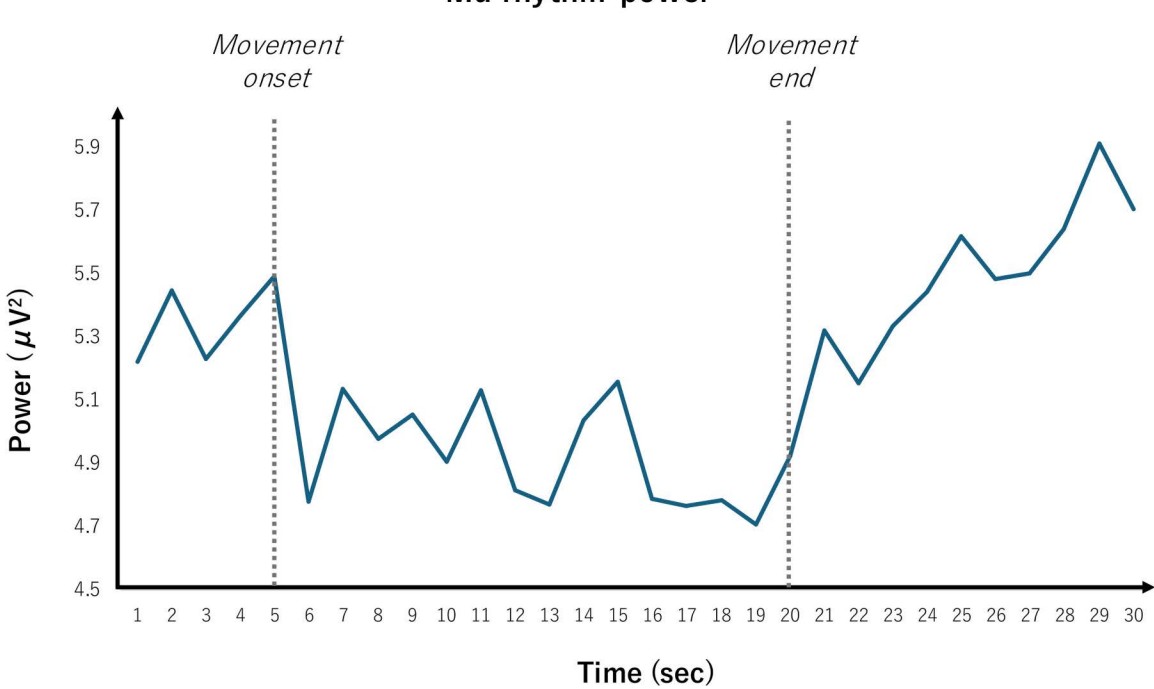

**Fig 3. The time course of the grand average for the mu rhythm power.** The mean Mu rhythm power of all participants was plotted for 30 seconds, starting 5 seconds before the right hand movement. The right hand movement continued for 15 seconds, after which participants were instructed to rest for 15 seconds with their eyes open. Each participant repeated this 8 times and the average Mu rhythm power for the 8 trials was calculated.

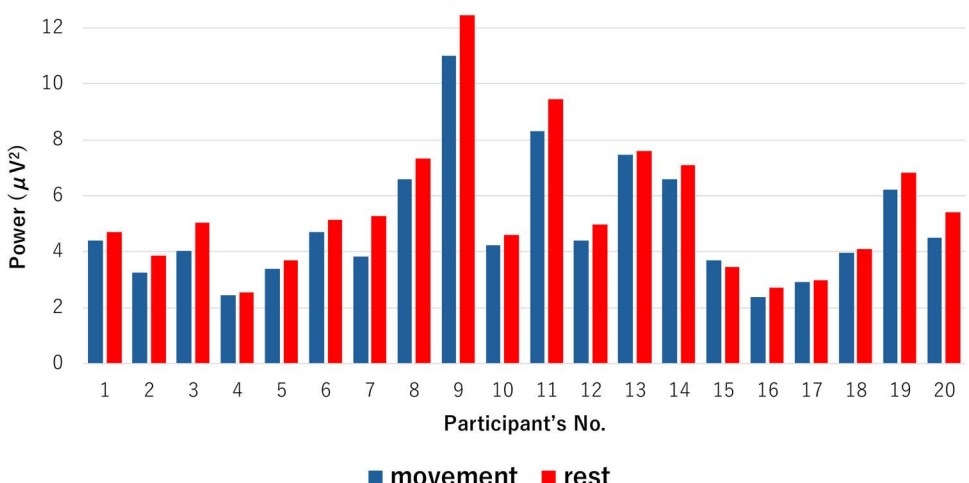

**Fig 4. Mu rhythm power for each participant.** The bar graph represents the mean mu rhythm power (μV²) for each participant in the right-hand movement condition and eye opened resting condition. The blue bar graph illustrates the mean mu rhythm power values in the right-hand movement condition, while the red bar graph depicts the mean mu rhythm power values in the eye opened resting condition.

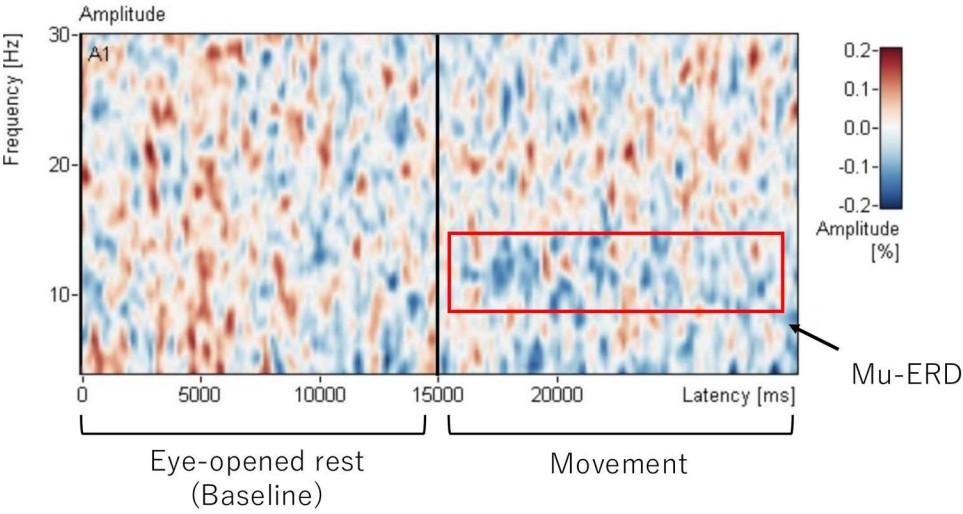

**Fig 5. The grand average of the amplitude-based time-frequency data for 15 seconds of eye-opened rest condition and the right-hand movement condition in 20 participants.** Amplitude higher than baseline at 15 seconds of eye opened resting was shown in red and lower activity in blue. Mu rhythm (8-13Hz) ERD compared to eye opened resting was showed during right hand movement (the right side).

suppression in the 9-12.5 Hz frequency band in the time-frequency data. These findings support the utility of ear EEG in capturing motor activity-related mu-ERD, are consistent with established research linking mu-ERD to sensorimotor cortex engagement during motor tasks [2,4,9,32,33], and demonstrate the potential of ear EEG in such applications.

The innovation of this study is the implementation of an ear EEG device, which is emerging as a practical tool for unobtrusive ambulatory monitoring of brain activity [12,14]. The

**Table 1. Results of the paired t-test comparing the time-frequency data of the right hand movement condition and the eye opened resting condition.**

| Cluster No. | Cluster value | Mean for movement | Mean for rest | Start Frequency | End Frequency | p-value |
|---|---|---|---|---|---|---|
| 1 | −20.6186 | 2.10309 | 2.23513 | 9 | 12.5 | p < 0.001 |
| 2 | −15.4775 | 2.43072 | 2.56981 | 4 | 6 | p = 0.001 |
| 3 | −2.30699 | 1.37657 | 1.42868 | 25 | 25.5 | p = 0.001 |

validity of ear EEG has been demonstrated through its use with common EEG paradigms, including alpha attenuation and auditory evoked potentials. It has been shown to accurately characterize the frequency spectrum and have a signal-to-noise ratio comparable to that of nearby scalp electrodes [13,19]. Previous studies suggest that ear EEG is comparable to conventional scalp EEG in terms of performance and accuracy for brain sources close to the ear, such as the temporal lobe [19,21,22]. In addition, several studies have shown a high correlation between the ear EEG and electrodes located in the forehead and inferior parietal [21,22]. These phenomena can possible to explained by volume conduction of electrical signals by structures other than the brain parenchyma, such as skin, skull, and cerebrospinal fluid [34,35]. Cerebrospinal fluid has high conductivity and negligible interindividual variability [36]. In our study, mu-ERD was measured with ear EEG, which may also be affected by volume conduction. Indeed, mu-ERD has also been shown to be measured in the lower part of the sensorimotor cortex and in the lower parietal lobes, compared to beta and gamma ERD [37,38]. Previous studies have demonstrated that mu-ERD is observed bilaterally during movement execution [39–41]. Consequently, we concluded that mu-ERD was observed during right-hand movement using this ear EEG device with only a 1-channel system. This research pioneers the detection of movement-related oscillatory activity via ear EEG. It is important to note that our study did not use conventional electroencephalography or other neuroimaging measures in combination, so it is impossible to confirm whether the study captured signals originating from the sensorimotor cortex. Nevertheless, the implications for neurorehabilitation are significant, particularly for the development of personalized wearable neurofeedback systems for motor impairments [9,10,23,42].

Interestingly, the time-frequency analysis revealed a significant decrease in the theta (4–6 Hz) and beta (25–25.5 Hz) frequency bands during active movement compared to a resting state. Such oscillatory changes in the alpha and beta bands, both before and after voluntary movements, are well documented [4,43–46]. In particular, beta oscillations are associated with motor improvements and have been proposed as markers of recovery in brain-computer interface therapies for post-stroke rehabilitation [47–49]. In addition, theta oscillations are often involved in brain activation during movement and are associated with cognitive functions related to the hippocampus and medial prefrontal cortex, affecting motor control, attention, and memory [50–55]. Ketenci et al. (2019) reported that the presence of alpha, beta, and theta rhythms provides superior monitoring of motor-related brain activity, highlighting the critical role of theta rhythm in movement initiation and execution [56]. Our results suggest that ear EEG may be able to detect these crucial oscillatory activities, highlighting its potential utility in the development of customized neurofeedback systems for rehabilitation purposes.

However, we must acknowledge the limitations of this study, such as the sample size and the exclusive focus on healthy individuals [57–59]. In addition, the absence of a control condition or comparison to a standard scalp EEG setup limits the ability to draw meaningful conclusions about the superiority or equivalence of ear EEG for detecting mu-ERD during hand movements. The ear EEG device was constrained by its capacity to measure only a single channel in this study. Consequently, the analysis of spatial information, such

as a topographical map or current source density, was not feasible. Comparative validation of ear electroencephalography with standard scalp electroencephalography has been done in previous studies [19,21,22]. Additionally, electromyographic signals were not recorded in this study. The timing of the start and end of the hand movement was confirmed by video recordings synchronized with the EEG. Future studies should expand the demographic scope, including patients with neurological disorders, to evaluate the clinical applicability of ear EEG in rehabilitation settings. Such efforts could provide important insights into the neural underpinnings of motor recovery and inform the development of more effective rehabilitation protocols.

## Conclusions

The purpose of this study was to evaluate the feasibility of the ear EEG device for monitoring mu-ERDERD during hand grasp and release movements. The statistical analysis showed a significant difference in mean mu rhythm power between the eye opened rest condition and the right hand movement condition and significant suppression in the 9-12.5 Hz frequency band in the time-frequency data. Our results suggest that ear EEG may be able to detect mu-ERD associated with hand movement, highlighting its potential utility in the development of neurofeedback systems for rehabilitation purposes. Ear EEG could be instrumental in refining rehabilitation strategies by providing in-situ assessment of motor function and tailored feedback.

## Acknowledgment

The authors would like to sincerely thank Mr. Kiichirou Arikawa, Mr. Ryosuke Sensui, and Mr. Yuto Doura of CyberneX Corporation for providing the ear EEG device and technical support.

## Author contributions

**Conceptualization:** Masaya Ueda, Ryouhei Ishii.

**Data curation:** Masaya Ueda, Keita Ueno, Takao Inoue, Misao Sakiyama, China Shiroma.

**Formal analysis:** Masaya Ueda, Keita Ueno.

**Funding acquisition:** Masaya Ueda.

**Investigation:** Masaya Ueda.

**Methodology:** Masaya Ueda, Ryouhei Ishii, Yasuo Naito.

**Project administration:** Yasuo Naito.

**Resources:** Masaya Ueda.

**Supervision:** Ryouhei Ishii, Yasuo Naito.

**Visualization:** Masaya Ueda.

**Writing – original draft:** Masaya Ueda.

**Writing – review & editing:** Keita Ueno, Takao Inoue, Misao Sakiyama, China Shiroma, Ryouhei Ishii, Yasuo Naito.

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
