## [Decision Letter · Decision Letter 0]

2 Jan 2025

PONE-D-24-31572Detection of motor-related mu rhythm desynchronization by ear EEGPLOS ONE

Dear Dr. Ueda,

Thank you for submitting your manuscript to PLOS ONE. After careful consideration, we feel that it has merit but does not fully meet PLOS ONE’s publication criteria as it currently stands. Therefore, we invite you to submit a revised version of the manuscript that addresses the points raised during the review process.

We look forward to receiving your revised manuscript.

Kind regards,

Fali Li

Academic Editor

PLOS ONE

Journal Requirements:

Initials of the authors who received each award: M.U

Grant numbers awarded to each author: 23K16581 and 22K21212

The full name of each funder: Japan Society for the Promotion of Science(JSPS)

URL of each funder website: https://www.jsps.go.jp  

Additional Editor Comments :

Please respond to the reviewers' comments.

Reviewers' comments:

Reviewer's Responses to Questions

**Comments to the Author**

1. Is the manuscript technically sound, and do the data support the conclusions?

Reviewer #1: Partly

Reviewer #2: Partly

2. Has the statistical analysis been performed appropriately and rigorously? 

Reviewer #1: Yes

Reviewer #2: No

3. Have the authors made all data underlying the findings in their manuscript fully available?

Reviewer #1: Yes

Reviewer #2: No

4. Is the manuscript presented in an intelligible fashion and written in standard English?

Reviewer #1: No

Reviewer #2: Yes

5. Review Comments to the Author

Reviewer #1: This study investigated the feasibility of an ear electroencephalography (EEG) device for monitoring mu-ERD during hand grasp and release movements. To do this, the authors calculated and compared mu rhythm power and time-frequency data from 20 healthy participants during right hand movement and eye opened resting. The experimental results show that there is a significant difference in the average mu rhythm power between the state of rest with open eyes and the state of right hand movement, and there is a significant inhibition in the 9-12.5Hz frequency band in the time-frequency data, which verifies the feasibility of using ear EEG equipment to monitor the ERD of mu rhythm during hand movement. My comments are as follows:

1) Abbreviation can have a list, and if only two words have an abbreviation, they don't have to be singled out.

2) Weird expression：

1. It looks like there's two sentences in there: Electroencephalography (EEG) is a leading method for noninvasive measurement

of human brain activity and has been used the use to track movement-related brain activity is well established.

2. EIt seems that ear EEG is becoming a promising new option, potentially providing a more subtle method for recording EEG signals that could enhance the feasibility of long-term observation of motorrelated brain activity in clinical settings[18–20].

3) The advantages of ear EEG in enhancing the spatial resolution of EEG were not elucidated, only the correlation of ear EEG with electrodes in the frontal and subparietal lobes was mentioned.

4) It is suggested to further verify the validity of ear EEG by adding a classification and comparison experiment on the EEG of grasping and relaxing tasks.

5) In addition to time-frequency analysis, it is suggested to analyze the effectiveness of the proposed method from several perspectives, such as topographic map topological distribution and cortex source estimation.

Reviewer #2: The current study aim to evaluate the feasibility of ear EEG device for monitoring ERD during hand movements, 20 subjects were included. The results obtained showed that the ear canal EEG electrodes were able to detect ERDs during hand movements, but the analysis of the results was not discussed in sufficient depth and suffered from theoretical shortcomings. My comments are as below:

1. The right hand movement task activates the left sensorimotor brain areas, i.e., ERD phenomena are observed mainly in the left side of the brain, and ERS phenomena are often observed ipsilaterally (right side of the brain). In the present study, the authors placed the recording electrodes in the right ear, but chose the right hand movement task for what reason? Were the ERDs observed correctly?

2. It is recommended that the authors present the results of the analyses for each subject, not just the pooled average results. In addition the variance of EEG features between trials for the same subject is also worth analyzing.

3. I don't think it is necessary to use the permutation method in the statistical analysis, it is sufficient to use paired t-test to directly compare the ERD values or power values during hand movement and resting.

4. Since this is a feasibility validation study, simultaneous acquisition of scalp EEG signals for comparison is recommended.

5. Table 1, start frequency 25.5, end frequency 25.5?

6. The resolution of the image is low.

7. The clarity and readability of the text can be further improved by refining the English language. I recommend the authors carefully review the manuscript for grammatical errors, and inconsistencies in word choice. Engaging a professional language editor or utilizing advanced proofreading tools could be beneficial in ensuring the text meets high linguistic standards.

6. PLOS authors have the option to publish the peer review history of their article (what does this mean? ). If published, this will include your full peer review and any attached files.

**Do you want your identity to be public for this peer review?** For information about this choice, including consent withdrawal, please see our Privacy Policy .

Reviewer #1: No

Reviewer #2: **Yes: ** Rui Zhang

---

## [Author Response · Author response to Decision Letter 1]

15 Jan 2025

Editors

PLOS ONE

Thank you for your letter and the reviewers’ comments concerning our manuscript titled Detection of motor-related mu rhythm desynchronization by ear EEG (ID: PONE-D-24-31572). These comments were valuable and helpful for revising and improving our paper. We have studied the comments carefully and have made the corrections. The main corrections in the manuscript and the responses are as follows.

RESPONSE TO REVIEWER 1:

Response: We wish to express our appreciation to the reviewers for their insightful comments, which have helped us improve the paper significantly. The revisions are highlighted in red font. Our responses to the comments are as follows.

1. Abbreviation can have a list, and if only two words have an abbreviation, they don't have to be singled out.

Response: Thank you for your suggestion. We have deleted the list of abbreviation.

2. Weird expression：

1. It looks like there's two sentences in there: Electroencephalography (EEG) is a leading method for noninvasive measurement

of human brain activity and has been used the use to track movement-related brain activity is well established.

2. EIt seems that ear EEG is becoming a promising new option, potentially providing a more subtle method for recording EEG signals that could enhance the feasibility of long-term observation of motorrelated brain activity in clinical settings[18–20].

Response: Thank you for your attention to detail. We revised as follows:

Page 4, Line 50-51: Electroencephalography (EEG) is a leading method for non-invasive measurement of human brain activity.

Page 4, Line 63-65: Ear EEG is a new approach that holds the potential to enhance the feasibility of long-term observation of brain activity in clinical settings[18-20].

3. The advantages of ear EEG in enhancing the spatial resolution of EEG were not elucidated, only the correlation of ear EEG with electrodes in the frontal and subparietal lobes was mentioned.

Response: Thank you for your comment. As you say, we have not mentioned the advantages of ear EEG in the spatial resolution. Previous studies verified the validity of signals measured using similar devices and conventional EEG equipment. This study aimed to verify whether it was possible to measure motor-related Event-related desynchronization of mu rhythm using an Ear EEG device that could only measure one channel. Therefore, we did not verify the spatial resolution in this study.

Please refer to the following sentence for information on the validity of the signals measured by the Ear EEG device and their relationship to conventional EEG devices.

Page 12, Line 236-243: The validity of ear EEG has been demonstrated through its use with common EEG paradigms, including alpha attenuation and auditory evoked potentials. It has been shown to accurately characterize the frequency spectrum and have a signal-to-noise ratio comparable to that of nearby scalp electrodes[13,19]. Previous studies suggest that ear EEG is comparable to conventional scalp EEG in terms of performance and accuracy for brain sources close to the ear, such as the temporal lobe[19,21,22]. In addition, several studies have shown a high correlation between the ear EEG and electrodes located in the forehead and inferior parietal[21,22].

4. It is suggested to further verify the validity of ear EEG by adding a classification and comparison experiment on the EEG of grasping and relaxing tasks.

Response: We agree with the reviewers' suggestion. This study has already compared the ear EEG data between grasping and resting with relaxed eye-open conditions. We used the resting eye-open condition as the control condition for two main reasons. First, we assumed that rehabilitation interventions basically would be carried out with the eyes open. The second is that the oscillatory activity in the 8 to 13 Hz frequency band (almost mu rhythm) clearly differs significantly between the eye opened and eye closed conditions.

5. In addition to time-frequency analysis, it is suggested to analyze the effectiveness of the proposed method from several perspectives, such as topographic map topological distribution and cortex source estimation.

Response: As the reviewer says, spatial analysis using data from multi-channel EEG, such as the topological distribution of topography maps and cortical source estimation, provides critical findings. However, we used the Ear EEG device, which could measure only one channel. Thus, we could not analyze spatial information such as a topo map or current source density. We added the following sentence to mention this limitation of this research.

Page 13-14 Line 275-277: The ear EEG device was constrained by its capacity to measure only a single channel in this study. Consequently, the analysis of spatial information, such as a topographical map or current source density, was not feasible.

RESPONSE TO REVIEWER 2:

Response: Thank you for reviewing our manuscript and offering valuable advice. We have addressed your comments with point-by-point responses and have revised the manuscript accordingly. We wish to express our appreciation to the reviewers for their insightful comments, which have helped us to significantly improve our manuscript. The revisions are highlighted in red font. Our responses to the comments are provided below.

1. The right hand movement task activates the left sensorimotor brain areas, i.e., ERD phenomena are observed mainly in the left side of the brain, and ERS phenomena are often observed ipsilaterally (right side of the brain). In the present study, the authors placed the recording electrodes in the right ear, but chose the right hand movement task for what reason? Were the ERDs observed correctly?

Response: Thanks for your insightful opinion. We realized that we may have lacked essential information. Several previous studies have shown that mu-ERD is observed ipsilaterally in the sensorimotor cortex before movement and bilaterally during movement execution. Therefore, we believe observing mu-ERD with either movement of the left or right hand is possible. We chose the right-hand movement because most of the subjects were right-handed. Previous studies show that handedness influences the proportion of pre-movement mu-ERD in the left and right peri-rolandic areas.

Page 7, Line 131-133: We chose the right-hand movement because most of the subjects were right-handed. Previous studies show that handedness influences the proportion of pre-movement mu-ERD in the left and right peri-rolandic areas[24,25].

Page 12-13, Line 249-251: Previous studies have demonstrated that mu-ERD is observed bilaterally during movement execution[39–41]. Consequently, we concluded that mu-ERD was observed during right-hand movement using this ear EEG device with only a 1-channel system.

2. It is recommended that the authors present the results of the analyses for each subject, not just the pooled average results. In addition the variance of EEG features between trials for the same subject is also worth analyzing.

Response: We agree that we should show the results of the analyses for each subject and the variance of EEG features between trials for the same subject. We added the figure and sentence below.

Page 10, Line 186-188: The time course of the grand average for the mu rhythm power is shown in Fig 3 and the mean mu rhythm power (μV2) for each participant in the right-hand movement condition and eye opened resting condition is shown Fig 4.

Page 10, Line 199-203: Fig 4. Mu rhythm power for each participant. The bar graph represents the mean mu rhythm power (μV2) for each participant in the right-hand movement condition and eye opened resting condition. The blue bar graph illustrates the mean mu rhythm power values in the right-hand movement condition, while the red bar graph depicts the mean mu rhythm power values in the eye opened resting condition.

3. I don't think it is necessary to use the permutation method in the statistical analysis, it is sufficient to use paired t-test to directly compare the ERD values or power values during hand movement and resting.

Response: Thank you for your careful review of the statistical analysis methodology. We agree with your suggestion. First of all, we would like to state that we have not performed the permutation method in the statistical analysis for comparison of the simple mu rhythm power during hand movement and rest. To compare the averaged mu rhythm power (8-13 Hz) of each participant between the right hand movement condition versus the eye open rest condition, Wilcoxon signed rank test analyses were performed. For the time-frequency data, we were analyzed by permutation test based on t-test and cluster analysis using BESA Statistics 2.1 (BESA GmbH, Germany). When comparing the differences in time-frequency data, a comparison is made for each voxel that is output according to the time-frequency resolution. Therefore, it is necessary to correct for multiplicity. We have added the sentence as below.

Page 9, Line 166-168: When comparing the differences in time-frequency data, a comparison is made for each voxel that is output according to the time-frequency resolution. Therefore, it is necessary to correct for multiplicity.

4. Since this is a feasibility validation study, simultaneous acquisition of scalp EEG signals for comparison is recommended.

Response: We agree that the simultaneous acquisition of scalp EEG signals for comparison is better. The objective of this study was not to make a comparison with conventional EEG; consequently, this data was not obtained. This is due to the fact that a significant number of ear EEGs, including those of the same type, have already been compared with conventional EEG. However, we have to realize this is the clear limitation of this study. Please refer to the following statement on research limitations.

Page 13, Line 272-275: In addition, the absence of a control condition or comparison to a standard scalp EEG setup limits the ability to draw meaningful conclusions about the superiority or equivalence of ear EEG for detecting mu-ERD during hand movements.

5. Table 1, start frequency 25.5, end frequency 25.5?

Response: Thank you for your careful pointing out of the table. We have mistaken to write. The correct starting frequency was 25 Hz. We have revised the table 1 and related text.

Page 11, Line 210-211: Interestingly, other significant reductions were also found at 4-6 Hz and 25-25.5 Hz (p = 0.001).

Page 13, Line 258-60: Interestingly, the time-frequency analysis revealed a significant decrease in the theta (4-6 Hz) and beta (25-25.5 Hz) frequency bands during active movement compared to a resting state.

6. The resolution of the image is low.

Response: We have increased the resolution of the images to a level that does not exceed the upload size limit. Please check the figures.

7. The clarity and readability of the text can be further improved by refining the English language. I recommend the authors carefully review the manuscript for grammatical errors, and inconsistencies in word choice. Engaging a professional language editor or utilizing advanced proofreading tools could be beneficial in ensuring the text meets high linguistic standards.

Response: We appreciate your suggestion. We utilized advanced proofreading tools and reviewed and revised our manuscript.

---

## [Decision Letter · Decision Letter 1]

3 Mar 2025

Detection of motor-related mu rhythm desynchronization by ear EEG

PONE-D-24-31572R1

Dear Dr. Ueda,

We’re pleased to inform you that your manuscript has been judged scientifically suitable for publication and will be formally accepted for publication once it meets all outstanding technical requirements.

Kind regards,

Fali Li

Academic Editor

PLOS ONE

Additional Editor Comments (optional):

Reviewers' comments:

Reviewer's Responses to Questions

**Comments to the Author**

1. If the authors have adequately addressed your comments raised in a previous round of review and you feel that this manuscript is now acceptable for publication, you may indicate that here to bypass the “Comments to the Author” section, enter your conflict of interest statement in the “Confidential to Editor” section, and submit your "Accept" recommendation.

Reviewer #1: All comments have been addressed

Reviewer #2: All comments have been addressed

2. Is the manuscript technically sound, and do the data support the conclusions?

Reviewer #1: Yes

Reviewer #2: Partly

3. Has the statistical analysis been performed appropriately and rigorously? 

Reviewer #1: Yes

Reviewer #2: Yes

4. Have the authors made all data underlying the findings in their manuscript fully available?

Reviewer #1: Yes

Reviewer #2: No

5. Is the manuscript presented in an intelligible fashion and written in standard English?

Reviewer #1: Yes

Reviewer #2: Yes

6. Review Comments to the Author

Reviewer #1: I appreciate the efforts made by the authors to improve the quality of this work. All my comments have been addressed.

Reviewer #2: (No Response)

7. PLOS authors have the option to publish the peer review history of their article (what does this mean? ). If published, this will include your full peer review and any attached files.

**Do you want your identity to be public for this peer review?** For information about this choice, including consent withdrawal, please see our Privacy Policy .

Reviewer #1: No

Reviewer #2: No

---

## [Editor Report · Acceptance letter]

PONE-D-24-31572R1

PLOS ONE

Dear Dr. Ueda,

I'm pleased to inform you that your manuscript has been deemed suitable for publication in PLOS ONE. Congratulations! Your manuscript is now being handed over to our production team.

Kind regards,

on behalf of

Dr. Fali Li

Academic Editor

PLOS ONE